

# Small scale variability of water vapor in the atmosphere: implications for inter-comparison of data from different measuring systems

Xavier Calbet[1], Cintia Carbajal Henken[2], Sergio DeSouza-Machado[3], Bomin Sun[4], and Tony Reale[5]

[1]AEMET, C/Leonardo Prieto Castro 8, Ciudad Universitaria, 28071 Madrid, Spain
[2]Institute of Meteorology, Feie Universität Belin (FUB), Carl-Heinrich-Weg 6-10, 12165 Berlin, Germany
[3]JCET, University of Maryland, Baltimore County, Baltimore, Maryland, USA
[4]IMSG, and NOAA/NESDIS/Center for Satellite Applications and Research, College Park, Maryland
[5]NOAA/NESDIS/Center for Satellite Applications and Research, College Park, Maryland

*Correspondence to:* Xavier Calbet
xcalbet@googlemail.com

**Abstract.** Water vapor concentration structures in the atmosphere are well approximated by Gaussian Random Fields at small scales ($\lesssim 6$ km). These Gaussian Random Fields have a spatial correlation in accordance with a structure function with a two-thirds slope, following the corresponding law from Kolmogorov's theory of turbulence. This is proven by showing that the structure function measured by several satellite instruments and radiosonde measurements do indeed follow the two-thirds law.

High spatial resolution retrievals of Total Column Water Vapor (TCWV) obtained from the Ocean and Land Color Instrument (OLCI) on board of the Sentinel-3 series of satellites qualitatively also show a Gaussian Random Field structure.

As a consequence, the atmosphere has an inherently stochastic component associated to the small scale water vapor features which, in turn, can make deterministic forecasting or Nowcasting difficult. These results can be useful in areas where a high resolution modeling of water vapor is required, such as the estimation of the water vapor variance within a region or when

searching for consistency between different water vapor measurements in neighboring locations.

## 1  Introduction

Meteorologists frequently need to determine the water vapor characteristics of an air parcel in the atmosphere. In the review from Wulfmeyer et al. (2015) it is shown how accurate and high resolution water vapor measurements are indispensable in the understanding and simulation of the water cycle. Unfortunately, it is common to have only partial information of these

atmospheric air parcels, usually coming from different measuring instruments or from a Numerical Weather Prediction (NWP) model. As an example, instruments on board satellites usually measure within a big spatial area whose radius is of the order of a few tens of kilometers. On the other hand, ground based stations or radiosondes measure an extremely small parcel of the atmosphere, which for all practical purposes can be considered as "point" measurements. Reconciling all these measurements, to make them consistent, is not an easy task and it is what constitutes the point to area problem (Loew et al., 2017). There are

many examples in which an adequate characterization of water vapor structure in the atmosphere is very useful or critical.


One such examples is passive satellite instruments measuring in the infrared or the microwave region of the spectrum. They measure over air regions of the order of linear measures of tens of kilometers (i.e. surfaces of few hundred square kilometers) and vertical thickness of a few kilometers. They actually measure an integration of the radiation coming from all the sub-parcels in which the measurement region can be subdivided. In fact, the variability of water vapor within the measurement region in the

field of view of satellite instruments can have significant effects when calculating the radiances impinging on the instrument via a Radiative Transfer Model (RTM). In this case, it is mandatory to know the variance of water vapor concentration within the remotely sensed air parcel (Calbet et al., 2018).

Another example is the calculation of instability indices for Nowcasting purposes, particularly the Convective Available Potential Energy (CAPE). Operational meteorology commonly has a first approximation of these instability indices coming

from NWP model forecasts or measured from remote sensing satellites. To refine such indices with ground based station measurements is not simple due to significant differences between different measurements or NWP models (Gartzke et al., 2017). This is caused by the high variability of water vapor in the atmosphere. To reconcile all estimates of instability indices, it is necessary to properly characterize the temperature and water vapor structure within the air parcel under study.

While it is common to treat atmospheric water vapor as a fluid in turbulent motion in study fields where measurements

are made at small scales, this is usually not so in other areas where larger scales are measured or modeled. Ground station or LIDAR measurements usually do apply concepts from turbulence theory (Lenschow et al., 2000; Wulfmeyer et al., 2010; Turner et al., 2014; Behrendt et al., 2015). But this is usualy not the case in RTM simulations (e.g. Sounders et al. (2018)) or NWP modelling (e.g. Milovac et al. (2015)). In this paper we will show how, for smaller scales below around 6 km, the atmosphere does indeed show a turbulent behavior. This scale length can be identified as the outer scale length of turbulence

quoted in the literature.

It should be noted that turbulent behavior in the atmosphere can be grouped in two different categories of scale lengths. One of them has a much smaller scale length than the phenomena studied here. These are, tipically, the scintillation measurements done at low zenith angles (e.g. Townsend (1965)) or turbulence measurements performed with LIDARs (e.g. Lenschow et al. (2000)), which usually quote an outer scale length of turbulence of a few tens to hundred of meters. The second category of

scale lengths, similar to the ones presented in this paper, deal with, for example, light propagation studies involving horizontal paths of propagation. They measure slow drifts of laser beams with amplitudes of a few arc seconds (Beckmann, 1965; Hodara, 1966; Lese, 1969) and are attributed to inhomogeneities in the atmosphere with scale lengths of 10 to 40 km (Zuev, 1982).

Achieving a complete characterization of water vapor concentration in an air parcel would require knowledge of the water vapor concentration at all "points" within such air parcel. As this goal is not achievable in practice, to bridge these gaps it

would be extremely convenient to have an approximate model of the behavior of water vapor concentration in the atmosphere at smaller scales. A way this is solved in other areas of geophysical sciences is by using kriging (Matheron, 1963). Kriging assumes that the average underlying fields follow what is mathematically known as a Gaussian Random Field (GRF) model. This assumption allows the computation of several characteristic parameters of the GRF, by which, practical conclusions on the expected behavior of the geospatial variable fields can be drawn (e.g. Chilès and Desassis (2018)).





As it turns out, the atmosphere is a fluid in turbulent motion, from which it follows that Kolmogorv's theory of turbulence applies. This theory basically states that fluids in turbulent motion have parameter fields which on average follow a GRF. In this paper it will be shown how the water vapor concentration in the atmosphere at smaller scales, on average, does indeed follow this pattern. This will be done in two ways. The first evidence will be to calculate what is known as the structure

function, that is, how the variances scale with distance. This will be done for several instruments and it will be shown that they do scale following the "two-thirds law" as expected from Kolmogorov's theory (Frisch, 1995). The second evidence will be to plot the fields at small scales and visually verify that they are indeed similar to a GRF. Overall, this modeling and characterization should prove very useful when trying to solve the above mentioned two problems, namely, finding consistency at small scales between different water vapor measurements in the atmosphere and modeling the fine scale behavior of water

vapor concentration for its application in RTM.

In section 2 the general theory concerning this study is presented. Section 3 presents the different data used in this paper. The methods used to analyze the data are shown in section 4. A discussion of the results is shown in section 5. Finally, a conclusion is presented in section 6.

## 2    Theoretical background

In this section the basic theory of turbulence is presented along with the mathematical definition of what a Gaussian Random Field is. The concept of structure function will also be introduced, alongside an example for the atmosphere. The theory is presented for two dimensional fields, but the atmosphere has in reality three dimensions. A few remarks regarding the third dimension are made in the last sub-section.

### 2.1    Kolmogorov's theory of turbulence

Kolmogorov's theory of turbulence is the set of hypotheses stating that a small-scale structure is statistically homogeneous, isotropic, and independent of the large scale structure. The source of energy at large scales is either velocity (wind) shear or convection. These set of hypotheses together with the Navier-Stokes equations are the foundations of Kolmogorov's theory. From these hypotheses, the experimentally observed "laws" can be derived. These are the two-thirds law and the law of finite energy dissipation (Frisch, 1995). Of these two laws, in this paper, we will only deal with the first one, the two-thirds law.

These two laws also imply that the fluid, at these scales, spatially follows a GRF.

A key concept in Kolmogorov's theory of turbulence is that of the structure function. This shows how the average of the squared difference of a fluid parameter between two spatially separated points behave as a function of their distance. Usually, a log-log plot is used to make this representation. This is illustrated in Fig 1. The panel in the center of this figure is from Calbet et al. (2018), where the structure function, obtained from radiosonde comparisons, is shown for temperature and water

vapor at different scales (for more details see also Section 3). Note that the temperature and water vapor structure functions have different units in the vertical axis. Temperature differences are shown in Kelvin and the ones for water vapor are a relative difference, hence with no units. This plot shows the usual behavior seen in fluids in the laboratory (e.g. Frisch (1995); Noullez





et al. (1997)). Namely, at small scales, known as the inertial range, the log of the variance of the difference grows linearly with the log of the distance with a two-third slope. This is what has been referred to above and is known as the "two-thirds law" (Frisch, 1995). At longer scales, the average of the differences squared diminish with distance, shown as the energy injection range in Fig. 1. At even longer scales, this average increases in what is here denominated the "synoptic range", in which the
differences increase with distance following a slope different from the inertial range. Here "synoptic range" is understood in a very relaxed sense ranging from a few tens to several thousand kilometers.

Apart from the energy injection range, which as we shall see later, is difficult to see with other instrumentation due to its narrow range, the conclusions that can be drawn from Fig 1 is that the atmosphere can be divided in two different ranges and behaviors. Above approximately 10 km on average we have what is called here the synoptic range. The atmospheric parameters
can be regarded as a more or less smooth field and it is what is usually reproduced by global NWP models. Below 6 km on average we have the inertial range, where the structure function follows the two-thirds law. In this range the water vapor field is extremely in-homogeneous and resembles a GRF.

It should also be noted that these considerations apply when averaging a big sample of measurements. In practice, when taking a smaller sample, the structure function will vary significantly from one region to another. One of such deviations from
the average structure function is the location in the vertical axis of the inertial range. If the data that fits the two-thirds law is high along this axis, it would indicate a high turbulence or concentration variability regime. If the data is placed in a lower place along the ordinate axis, it would nominally indicate a lower turbulence or concentration variability. The exact position where these points lie in the structure function graph will depend on the degree of turbulence of the region being analyzed. Another deviation from the average structure function is the frontier between the inertial and the synoptic range which can, as
we shall see later, vary significantly from one region to another. But, for the atmosphere, on average, this frontier lies around 6 to 10 km.

Kolmogorov's theory also implies that in the inertial range the parameter under study follows a GRF on average. To have a feeling of how a GRF looks like, the very left image in Fig. 1 shows a synthetically generated one, explained in more detail below. This is the behavior we can expect from the parameter field at small scales, which can be useful in obtaining further
conclusions from the measurements.

## 2.2  Gaussian Random Fields (GRF)

In this paper the only parameter we will focus on is one scalar field, namely the atmospheric water vapor concentration. This study could be easily extended to more parameters, like adding temperature. A field is defined by

$$f(\mathbf{x}), \mathbf{x} \in \mathbb{R}^2, \tag{1}$$

where $\mathbf{x}$ is the position in space which constitute the two horizontal dimensions. The third, vertical dimension will be dealt with in the following section.

A random field is one in which the value of the parameter is random and follows a certain probability distribution. In the case of Gaussian random fields (e.g. Rue and Held (2005)), if we have the value of a field, $f$, in position $\mathbf{x_1}$ and another one in





position $\mathbf{x_2}$, these random values will follow a multivariate Gaussian distribution such that the expected value of the field is

$$m(\mathbf{x}) \equiv\, <f(\mathbf{x})> \tag{2}$$

and their covariance is

$$C(\mathbf{x_1},\mathbf{x_2}) \equiv\, < [f(\mathbf{x_1}) - m(\mathbf{x_1})] [f(\mathbf{x_2}) - m(\mathbf{x_2})] >, \tag{3}$$

where the symbols $<>$ denote the expected values of the parameters enclosed within them. It is also usually assumed, as will be done here, that the field is stationary, meaning the mean is constant,

$$m(\mathbf{x}) = \text{constant}. \tag{4}$$

Another condition that is often used is that it is also homogeneous and isotropic, meaning that the covariance is only a function of the Euclidean distance,

$$C(\mathbf{x_1},\mathbf{x_2}) = C(|\mathbf{x_1} - \mathbf{x_2}|). \tag{5}$$

The structure function is defined as the relation between the expected value of the squared field difference between two points versus their distance. If it is further assumed that the fluid is in the inertial range where Kolmogorov's theory applies, then, defining the distance between two points as

$$r \equiv |\mathbf{x_1} - \mathbf{x_2}| \tag{6}$$

the structure function, defined as $S(r)$, will follow the two-thirds law

$$S(r) \equiv\, < [f(\mathbf{x_1}) - f(\mathbf{x_2})]^2 >\, = K r^{2/3}. \tag{7}$$

Note that $K$ is a measure of the variability of the field, or equivalently, provides a measure of where the curve is placed in the vertical axis of the structure function plot.

The notation of the covariance can now also be simplified to

$$C(r) \equiv C(|\mathbf{x_1} - \mathbf{x_2}|). \tag{8}$$

The relationship between the structure function and the covariance will be needed in a later stage. Some simple algebra shows that

$$S(r) = 2C(0) - 2C(r), \tag{9}$$

which implies that the structure function has a similar algebraic behavior as the covariances.

In the real atmosphere, all this is, of course, a simplification. Nevertheless, we will show that this approximation holds relatively well for small scales of observed water vapor concentrations. In summary, a Gaussian Random Field (GRF) will be understood in this paper as a random field satisfying Eqs. 2, 3, 4 and 7.





## 2.3 Consideration of the vertical dimension

In this sub-section a very brief mention in how the vertical dimension can be dealt with will be given. These considerations are particularly important for satellite remote sensing in the thermal infrared spectral region.

The vertical dimension could easily be accounted for if it is added to the spatial coordinate variable $\mathbf{x}$. In meteorology and
satellite remote sensing it is convenient to divide the atmosphere into vertical layers. Satellites, in particular, can be considered as instruments observing the atmosphere divided in several layers of finite thickness. For example, in the thermal infrared, to a very rough first approximation, for any one spectral channel the measured satellite radiance can be considered as an average emittance over several layers. Therefore, the particulars of the structure function observed by the satellite will depend on the vertical correlation between these layers. If the layers have completely independent statistical properties, then the covariances
of each layer can be averaged to give the combined covariance, with similar consequences for the structure function due to Equation 9. This is the typical $\sqrt{N}$ covariance diminishing factor of the average of a random variable. If, on the other hand, the layers have a perfect vertical correlation, then the covariances of all the layers combined will be equal to the covariance of one layer and the structure function will also follow this behavior.

In summary, if the vertical layers are statistically independent, the satellite observed structure function will have a much
lower $K$ value (or covariance) than each individual layer. The physical reason behind this is that the effect averages out as the different layers contribute in completely independent ways. If, on the other hand, the layers have a perfect vertical correlation, then the structure function of the combined layers will be the same as the one for an individual layer. This can be conceptually idealized as if we had identical values for all layers. In other words, for the GRFs to be visible from satellite observations some form of vertical correlation between atmospheric layers is needed to have an observable effect. This will come naturally
in some specific meteorological cases such as the presence of water vapor rolls which do show a high vertical correlation (Carbajal Henken et al., 2015).

## 3 Data

In this paper, one NWP model (ECMWF) and several satellite and radiosonde instrument data has been used. The data is detailed in the following subsections.

### 3.1 Radiosonde data

The data from radiosonde measurements from the EUMETSAT EPS/MetOp campaigns made in 2007 and 2008 at Lindenberg (Germany) and Sodankylä (Finland) observatories has been used. In these campaigns, two consecutive radiosondes were launched from the same site separated by about 50 min. in time. These type of measurements are usually referred to as sequential sondes. The instrument payload analyzed here are the conventional RS92 radiosondes. The data has been processed by
GRUAN (Dirksen et al., 2014), which, among other advantages, greatly removes the humidity measurement dry biases usually present in RS92 measurements at the high troposphere (Miloshevich et al., 2009).





Sonde measurements sample the atmosphere every second as the radiosonde ascends in the air. This effectively means measuring the troposphere in layers around 0.6 to 0.1 hPa thick in the pressure levels used in this study, which range from 950 to 200 hPa respectively. The humidity measurements from the first radiosonde of the sequential sonde pair is vertically interpolated to the vertical pressure grid of the second sonde. By doing this, both water vapor measurements from each of the

pair of sequential sondes can be compared directly. To calculate the structure function, the normalized differences between water vapor partial pressure measurements from each radiosonde at the same pressure level are calculated. If $e_1$ and $e_2$ are the partial water vapor pressure for the first and second sonde respectively, then the normalized difference is calculated with

$$\delta e/e \equiv \frac{(e_2 - e_1)}{(e_1 + e_2)/2}. \tag{10}$$

Note that this quantity has no units.

For temperature, the temperature difference is directly calculated,

$$\delta T \equiv T_2 - T_1, \tag{11}$$

where $T_1$ and $T_2$ are the temperatures of the first and the second radiosonde respectively. Its corresponding unit is Kelvin. To these differences, an effective distance is assigned. This effective distance is the real spatial distance between sequential sondes plus the time difference multiplied by the wind speed measured by the radiosondes at that level. The average of the square of

this normalized water vapor concentration or temperature difference is then calculated for different effective distance bins. In order to achieve a significant sample size, results from all sample pairs and for all radiosonde pressure levels between 950 and 200 hPa have been combined. The resulting total number of data pairs, coming from the 625 sequential sonde pairs, is 658,217. From these calculations, the structure function for both temperature and water vapor can be plotted. This is shown in Fig. 1. For more details on this see Calbet et al. (2018).

**3.2 SEVIRI on board MSG**

The Spinning Enhanced Visible Infrared Imager (SEVIRI) is an imager instrument on board of the Meteosat Third Generation (MSG) geostationary satellite (Schmetz et al., 2002). SEVIRI is a 50 cm diameter aperture, line by line scanning radiometer. It provides image data in four visible and near infrared channels and eight infrared channels. The only channel used in this paper is the one centered in 6.25 $\mu$m with a spectral interval, in which 99% of the energy is detected, between 5.35 and 7.15 $\mu$m and

a $NE\Delta T = 0.75$K @ 250K. Its spatial resolution (sampling distance) is 3 km at sub-satellite point. The SEVIRI instrument measures the complete observable disk from geostationary orbit with a repeat cycle of 15 min. It yields an image, for the 6.25 $\mu$m channel, of 3712 times 3712 pixels covering the complete disk for each repeat cycle. The images used are the Level 1.5 ones, which are radiometrically calibrated and geolocated. They have been obtained via the EUMETSAT archive in HRIT format.

The SEVIRI image date and time for the determination of the structure function has been selected randomly and corresponds to 20/08/2021 at 10:00Z. The corresponding "Airmass RGB" image can be seen in Fig. 2. To avoid high satellite zenith angles effects in the radiative transfer, a region centered in the sub-satellite point has been selected. The region is a square of $1000 \times 1000$ pixels centered at nadir (roughly an area of $3000 \times 3000 \text{ km}^2$). It is highlighted in Fig. 2 as a red square.





The 6.25 $\mu$m channel is centered, from the spectroscopic point of view, in the highly absorptive portion of the water vapor band. It is therefore a channel that mainly detects water vapor in the high troposphere. Because of this, it is classified as a "water vapor" channel. It is almost completely insensitive to surface effects such as skin temperature or surface emissivities, simplifying its radiative transfer modeling. For all these reasons it is used to detect water vapor in the high troposphere, a

magnitude that will be denoted as High Level column Water Vapor (HLWV) in this paper. The HLWV will be the integral over a column of the water vapor content between 500 and 0 hPa and will have units of millimeters of water, which we denote as mm.

The structure function could be determined directly from the measured radiances, but these do not constitute an atmospheric parameter. It is therefore best to convert these radiances into a measurement of water vapor such as the HLWV defined above.

Only a first order approximation of the HLWV is needed, since the structure function results are fairly robust to any such estimation. To estimate the HLWV from the 6.25 $\mu$m channel radiances we start by using an atmospheric profile representing all other atmospheric profiles in the selected region. The profile has been chosen to be inside the selected region and sufficiently far away from any high level cloud. Its location is shown as a cyan dot in Fig. 2. The temperature and water vapor values of the profile are obtained from an NWP forecast model of the region. In particular, it is an ECMWF forecast from an analysis

on 20/08/2021 at 00Z and a forecast step of 10 hours. The top of atmosphere radiances are calculated for this profile using the RTTOV radiative transfer model (Sounders et al., 2018). The water vapor content of this profile is perturbed at all levels and with varying satellite zenith angles such that the radiance dependency versus the HLWV is obtained. After correcting for the satellite zenith dependency, a linear relationship can be fitted between the observed radiance and the HLWV, with the final result being

$$HLWV \cong -0.32894474\, R_{6.25\,\mu\mathrm{m}} \cos(\theta)^{-0.415} + 3.08690795, \tag{12}$$

where $R_{6.25\,\mu\mathrm{m}}$ is the measured radiance in $\mathrm{mW/m^2/sr/cm^{-1}}$, $\theta$ is the satellite zenith angle and the output units for $HLWV$ is mm.

The RTM requires that only scenes unaffected by clouds are analyzed. Since the 6.25 $\mu$m channel mainly sees the upper troposphere, scenes with any mid or high clouds need to be excluded. To to this, the Nowcasting Satellite Application Facility

(NWC SAF) software package (NWCSAF , 2018) is used. It is executed on this region to determine the cloud types. The Cloud Type (CT) is a product generated by the NWC SAF software that provides the could types in a categorical variable. Only pixels with scenes that are clear or with very low level clouds are selected for the analysis.

### 3.3 OLCI on board Copernicus Sentinel-3

The Ocean and Land Color Instrument (OLCI) is a push broom imaging spectrometer that measures solar radiation reflected

by the Earth. OLCI is on board of the polar sun-synchronous Sentinel-3 Earth observation satellite series dedicated to ocean and land observation (Donlon et al., 2012). OLCI measures in a swath width of 1270 km with a ground spatial resolution of 300 m in 21 spectral bands between 400 and 1020 nm. Their radiometric accuracy is of $0.1\%$.





From this instrument a retrieval of Total Column Water Vapor (TCWV) can be performed. The method used in this paper is based on the Copernicus Sentinel-3 OLCI Water Vapor (COWa) product. It uses an optimal estimation method to retrieve TCWV from the Oa17, Oa18, Oa19 and Oa20 OLCI bands. Because of this, it provides both a measurement of the TCWV and its uncertainty. The properties of the OLCI channels allow for accurate determination of the TCWV over land in clear sky scenes. The TCWV estimation over ocean is far more uncertain and is not used in this paper. The method is fully described in (Preusker, Carbajal Henken and Fischer, 2021).

A slot on 31/08/2016 at 09:45Z covering southeast Germany and western Czech Republic has been selected. This region is located over land and consists mostly of clear sky scenes making it an ideal candidate for the accurate measurement of TCWV with the OLCI instrument. The TCWV for this field is represented in Fig. 3. The high variability of water vapor can be clearly spotted in this image as has been recognized before (Carbajal Henken et al., 2015). As explained above, each pixel has its own precise value of uncertainty, however, to have a mental picture, we can keep in mind that the uncertainty in this region is around $0.33\,\mathrm{mm}$.

### 3.4 ECMWF NWP model

A comparison of the measured structure functions and small scale variability with other sources can be instructive. For this reason, the same region as the one selected for OLCI is also selected for an NWP model. The NWP model is the ECMWF operational global model retrieved from ECMWF's archive. The model is obtained with a regular latitude/longitude grid of $0.125° \times 0.125°$. Only the TCWV parameter is obtained from the model, which is is a forecast predicted from an analysis on 31/08/2016 at 00Z with a step of 10 hours, which implies a validity time of 10:00Z on that same day. The field is plotted in Fig. 4. The blue line indicates the contour of the corresponding OLCI observation from Fig. 3. A quick visual comparison of OLCI's and ECMWF's TCWV fields (Figs. 3 and 4) show that they both share similar structures. The ECMWF field shows a lower spatial resolution and smaller contrast indicating a lower TCWV range with respect to OLCI.

### 4 Methods

In this section the methods applied to the data are discussed. In a first sub-section, the way to calculate the structure function from the data is explained. Two different types of structure functions are used. The first one is denominated "pixel centered structure function". It is a structure function that is calculated on each and every pixel of the image. The second one is an "average structure function" and, as it names indicates, it is an structure function calculated by averaging many pixel centered structure functions.

In a second sub-section, the calculations to derive the typical mathematical properties of GRFs are explained. In particular, a histogram is obtained from the measurements, which should follow a Gaussian distribution. Also several synthetic GRFs are generated, which are later compared to the measurements.



### 4.1 Water Vapor Structure Function

The water vapor structure function is calculated from the data in two different ways. One of them is centered in a particular pixel of field or image, which will be called "pixel centered structure function". The second one of them is an average over the whole satellite image or NWP field, which will be denoted by "average structure function". The way to calculate them is
described below.

#### 4.1.1 Pixel centered structure function

The goal is to have a structure function centered on each an every pixel within the satellite image or NWP field. Depending on the source of the data, the water vapor structure function is calculated for different parameters: TCWV for OLCI and ECMWF, HLWV for MSG. Note that for radiosondes (Fig. 1) the partial water vapor pressure was used. To simplify the notation, any
of these water vapor parameters will be denoted by the variable $w$. The pixel where the structure function is calculated will be denoted as the "central pixel" regardless of its location within the image or field. The very first step is to divide the total distance range, from the minimum observable by the instrument to a maximum of 100 km, in several discrete bins. Then, the relative difference between the parameter in the central pixel and all the surrounding ones up to a distance of 100 km is calculated. Similarly to what was done for sondes in Eq. 10, the relative difference is defined as the difference of the parameter divided by
the average. So, if $w_0$ is the water vapor parameter located in the central pixel and $w_1$ is the water vapor parameter of a pixel at a certain distance from the central one, then the relative difference is defined as

$$\delta w/w \equiv \frac{w_1 - w_0}{(w_0 + w_1)/2}. \tag{13}$$

Note that in the case of satellite measurements or NWP fields used in this paper, the measure of water vapor ($w$) is a columnar amount (TCWV or HLWV) while for the sondes it was a "point" measurement of partial pressure of water vapor ($e$) at a certain
pressure.

After this, the distance between these two points is calculated. The square of the relative difference is calculated and its value is accumulated into its corresponding distance bin. A record of the number of occurrences in each bin is kept. To achieve relevant statistics, specially for the short distance ranges, the number of cases needs to be increased. This is done by shifting the pixel where the origin of distances is located around a $5 \times 5$ pixels square surrounding the central pixel. In other words, the
exact same calculation is repeated within a $5 \times 5$ pixel square surrounding the central pixel. Once all the points are processed, the average is calculated by dividing the accumulation by the number of occurrences in each distance bin. The uncertainty of this average is also estimated statistically. In the end, the average of the relative difference squared, $< (\delta w/w)^2 >$, together with its uncertainty, as a function of distance is obtained. To obtain the structure function, the logarithm in base 10 of these two quantities can be plotted. Examples of pixel centered structure functions can be seen in Figs. 5 and 6. Note the different
y-intercepts when comparing these figures due to the different in-homogeneities or turbulence intensities. The white regions in these figures correspond to pixels where the potential presence of clouds has been detected and have, therefore, been masked out. These kind of figures could be reproduced for any other pixel on the complete OLCI field, showing similar results.





### 4.1.2 Average structure function

To reduce the uncertainties in the structure function or to have a global picture of it, it is convenient to obtain an average of all the pixel centered structure functions of a given instrument. This is done in three steps.

1. The first step is to bring all cases to the same vertical axis in the structure function. This is achieved by averaging the structure value, $\log_{10}(< (\delta w/w)^2 >)$, for all distances smaller than a threshold. This threshold distance is the maximum distance in which the structure function still approximately retains the two-thirds slope for all the pixel centered structure functions of a given instrument. Defining this threshold as $r_{lim}$, this average, $a_i$, can be defined in mathematical form as

$$a_i \equiv < \log_{10}[< (\delta w(r_i)/w)^2 >] >: r_i < r_{lim}, \tag{14}$$

where the sub-index $i$ labels a particular central pixel in the pixel centered structure function and emphasizes that there will be the same number of $a_i$ as pixels in the image or field. $r_i$ denotes the distance of any other pixel to the central one. The average of this parameter, denoted by $a$, will also be important

$$a \equiv < a_i > . \tag{15}$$

Because of their different spatial resolutions, the values of $r_{lim}$ will depend on the instrument or model used. Table 1 summarizes the values used for different instruments or model. These values have been derived by verifying empirically that below this distance the pixel centered structure function retains approximately the two-thirds slope for all pixels. The NWP model data (ECMWF) is exceptional because it lacks the very small scales, and the $r_{lim}$ value in Table 1 is one where a meaningful average can be obtained.

2. All components of the structure function are normalized by the $a_i$ factor, taking into account the logarithm, such that they are all brought to the same level,

$$\frac{(\delta w(r_i)/w)^2}{10^{a_i}}. \tag{16}$$

3. These re-scaled structure function components are now averaged and brought back to the average level

$$\left\langle \frac{(\delta w(r_i)/w)^2}{10^{a_i}} \right\rangle 10^a. \tag{17}$$

4. Finally, the logarithm of this quantity is taken to have the final value for the average structure function

$$\log_{10}\left[ \left\langle \frac{(\delta w(r_i)/w)^2}{10^{a_i}} \right\rangle 10^a \right]. \tag{18}$$

Examples of various average structure function are shown in Fig. 7.

## 4.2 Gaussian Random Fields

To demonstrate that water vapor structures at small scales do resemble GRFs first we must verify that measurements on an individual pixel do follow a Gaussian distribution by looking at its histogram. As a second step, water vapor measurements must visually resemble a GRF. For this, two synthetically generated GRFs have been created which can be compared to a
spatial zoom into a OLCI TCWV measurement region. How these plots have been generated is described below.

### 4.2.1 Gaussian Histogram

A square of $60 \times 60$ pixels (roughly $18 \times 18$ km) is selected from the OLCI TCWV measurements. The TCWV values of all the pixels in this square is subtracted from the average TCWV. This is repeated for all other similar square areas over the complete OLCI measurement region without any overlap between the squares. All these TCWV differences, $\delta\,TCWV$, are
collected together to generate a normalized histogram. This is shown in Fig. 8. Also plotted is an overlayed Gaussian function with standard deviation equal to the one obtained from the data ($0.899\,\mathrm{mm}$).

### 4.2.2 Synthetic GRFs

To get a feeling of the product noise or uncertainty a GRF with its standard deviation equal to the average OLCI COWa TCWV uncertainty (around $0.33\,\mathrm{mm}$) and also with no spatial correlation ($C(r) = 0$) is plotted. This is represented in the left image
of Fig. 9.
    To generate a synthetic GRF which follows the two-thirds law, an algorithm following the isotropic spectral method (Paludo et al., 2015) has been selected. The result can be seen in the central image of Fig. 9.

### 4.2.3 Measured OLCI TCWV fields

To appreciate the small scale features of the OLCI COWa TCWV fields, a zoom has been performed in a randomly selected
region centered on $(lon, lat) = (12.2344°, 49.6135°)$ and covering an area of about $6 \times 6\ \mathrm{km}^2$. This is shown in the right image of Fig. 9. Any other region selected would show the same structure.

## 5 Results and discussion

In this section, the results are shown. First, the structure functions will be discussed and, in a later section, qualitative and quantitative views of the GRFs will be shown.

## 5.1 Structure functions

The average structure function for several meteorological satellites and the ECMWF NWP model are shown in Fig. 7. Also shown is the plain structure function from radiosondes (green curve) obtained from Calbet et al. (2018). Although they all constitute structure functions for relative differences of water vapor, in these plots we emphasize that different water vapor





concentrations, pressure levels, spatial regions and validity times have been used. The radiosonde structure function is calculated from the partial pressure of water vapor and spans pressure levels from 950 to 200 hPa. The MSG structure function has been derived from the column water vapor from 500 to 0 hPa, which is referred to as HLWV in this paper. The OLCI and ECMWF structure functions are taken for very similar regions and validity times, and also using the same TCWV water vapor

parameter, so they should be very comparable.

Fig. 7 clearly demonstrates how all structure functions quite precisely follow a two-thirds law for spatial scales smaller than approximately 6 km. This is highlighted by the linear fit shown in this same figure and labeled as LF. The slope of the fitted linear regression is also shown in the inset together with its uncertainty. A linear fit with an exact two-thirds slope is also shown as the dotted black line for comparison. All linear fit slopes do show a two-thirds value within their uncertainty range.

Another distinct feature of this figure is the wide range of variability of water vapor, i.e., the different displacement of the curves in the vertical axis. The radiosonde data is measuring in very thin layers of at most 0.6 hPa in depth. All the other instruments measure in extremely thick layers or even the complete atmospheric column. This implies that the diminishing of the variance (smaller $K$ values) as the vertical integration range increases might play a significant role here (see Section 2.3).

Even though they are indeed measuring the same parameter at the same location in space and time, the contrast between

the OLCI and ECMWF NWP curve is significant. Since the ECMWF NWP is a global model it lacks the information at smaller scales than 6 km. Also, the variability of the ECMWF model is significantly smaller than the one from OLCI. This can also be verified visually comparing the direct products from Figs. 3 and 4. The reason for this is unknown. It could be a low representativity of the ECMWF NWP model or it could be caused by the NWP model trying to average the water vapor over the small scales due to its horizontal scale and the way it handles the water vapor. The consequence of this is clear, a global

NWP model might have a lower variability than other higher resolution measurements.

Finally, a feature that also stands out is the presence of a small region in which the variability decreases with distance, present in the OLCI pixel centered structure functions (Figs. 5 and 6) and the sonde structure function (Fig. 1, labeled as "energy injection range", or the green curve in Fig. 7). This is clearly absent for the average structure functions from MSG, OLCI and ECMWF in Fig. 7.

The pixel centered structure function obtained from OLCI, of which two are shown in Figs. 5 and 6, are also interesting. It can be verified that the variability also changes significantly from one region to another, as seen by the vertical location of the structure functions shown in the left plots of these figures. This variability difference spans almost an order of magnitude between these two regions. The inertial range, in which the structure function follows a two-thirds law, also has different sizes depending on the region. The region in Fig. 6 has a slightly bigger inertial range size ($\sim 1.5\,\mathrm{km}$) than the one in Fig. 5

($\sim 1.8\,\mathrm{km}$) and both of them are well below the average value of 6 km. In these pixel centered structure functions, the inertial range does not closely follow a two-thirds law as opposed to the averaged structure functions seen before. In particular, Fig. 6 shows a fairly "noisy" structure function in the inertial range and its slope is quite far away ($0.90 \pm 0.14$) from the ideal two-thirds value. Both of them show a range in which the variability decreases with distance, the "energy injection range", mentioned above.



## 5.2 Gaussian Random Fields

The first property for a random field to be Gaussian is that individual pixels must follow a normal distribution. This is verified in the histogram plotted in Fig. 8, which closely follows a Gaussian distribution.

To show that the OLCI TCWV does indeed behave like a GRF, several different panels are represented in Fig. 9. In the right panel of Fig. 9 the actual measured TCWV difference field has been plotted. More specifically, the difference with respect to the average of the field is shown, which is labeled as $\delta\,TCWV$ in the figure. This precise region is a zoom of the image shown in Fig. 3 centered at $(lon, lat) = (12.2344°, 49.6135°)$. Although a particular region of the OLCI measurements has been selected for the right image of Fig. 9, any other region within the complete field (Fig. 3) does share the same general aspect at these small scales.

On the left panel, a somehow especially restricted GRF has been generated. Its particularity is that it has no spatial correlation ($C(r) = 0$) and the standard deviation of its normal distribution is set to $\sigma = 0.33$mm. This value is approximately the uncertainty of the OLCI COWa TCWV retrieval fields used throughout this paper. This is exactly how the measurements would look like if only "retrieval noise" would be present. The measured TCWV (right panel) shows a very different behavior. The measurements show a spatial correlation not present at all in this synthetic field and also the variability is much higher than the simulated counterpart.

The measured TCWV can be compared with a synthetically generated GRF that does follow a spatial correlation following the two-thirds law. This is shown in the central panel of Fig. 9. The general aspect is remarkably identical to the measurement. This constitutes further proof that water vapor concentration in the atmosphere does indeed resemble a GRF at small scales.

## 6 Conclusions

The average structure function follows quite accurately a two-thirds law at small scales for several instruments (Fig. 7). This implies that the atmosphere, on average and at these scales, is following Kolmogorov's theory of turbulence. As a consequence, water vapor fields can be approximated as Gaussian Random Fields (GRF) at these scales. To confirm this, the histogram of individual pixels is shown to follow a Gaussian distribution (Fig. 8). Furthermore, the direct high resolution measurements of water vapor in the atmosphere (right panel of Fig. 9) show clear similarities with synthetically generated GRFs with similar spatial properties (central panel of Fig.9).

These assumptions can be applied only to scales below approximately 6 km on average. They do not apply at scales greater than this value, since the structure function deviates from the two-thirds law (Fig. 7) and the water vapor fields leave the GRF variability and start showing a "synoptic" structure (Figs. 3 and 4). This typical scale approximately coincides with other similar studies made with GPS phase difference observations (Kermarrec and Schön, 2020), which estimate an outer scale of turbulence of around 4 km, or with modeled atmospheric turbulence simulation, which show an outer scale length for potential temperature of about 10 km (Fig. 1 from Tung and Orlando (2003)).

As a consequence of these results, the water vapor concentration in the atmosphere is inherently turbulent and chaotic at these small scales. There will always be a random component which will be impossible to measure on a full spatial scale in



general, and, in particular, that of a typical 15 km satellite infrared sounder footprint. Water vapor near the surface is also a critical parameter for the triggering of convection. This means that Nowcasting will always have an inherently stochastic component associated to it. Because of this, it is highly likely that the best approach in making forecasts for Nowcasting would be to have a probabilistic approach to them.

5      Global NWP models do not seem be able to accurately reproduce the intensity of the variability of water vapor at small scales (Fig. 7). They also do not yet have the spatial resolution to resolve them. It is therefore imperative to use other methods to estimate this small scale variability of water vapor. Measurements or NWP models with a high spatial resolution could be a practical solution if these kind of features are needed.

      All these results can be of practical importance to estimate the variability of water vapor within a region. It can also be 10 applied in making several neighboring measurements consistent, since the random differences between both measurements can be estimated.

*Acknowledgements.* We thank Miguel Angel Martínez Rubio for assisting in the conversion of SEVIRI/MSG radiances into column water vapor content.





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





**Table 1.** Values of $r_{lim}$ for different instruments or model.

| Instrument | $r_{lim}$(km) |
| --- | --- |
| MSG/SEVIRI | $10^{0.8} = 6.31$ |
| Sentinel-3/OLCI | $10^{0} = 1.00$ |
| ECMWF | $10^{1.3} = 19.95$ |

Paludo, L., Bouvier, V., Corrêa, L., Cottereau, R., Clouteau, D.: Efficient Parallel Generation of Random Field of Mechanical Properties for Geophysical Application, Proc. 6th International Conference on Earthquake Geotechnical Engineering, Christchurch, New Zealand, 1-4 November 2015.

Miloshevich, L. M., Vömel, H., Whiteman, D., and Leblanc, T.: Accuracy assessment and correction of Vaisala RS92 radiosonde water vapor
measurements, J. Geophys. Res.-Atmos., 114, D11305, doi:10.1029/2008JD011565, 2009.

Milovac, J., K. Warrach-Sagi, A. Behrendt, F. Späth, J. Ingwersen, and V. Wulfmeyer: Investigation of PBL schemes combining the WRF model simulations with scanning water vapor differential absorption lidar measurements, J. Geophys. Res. Atmos., 121, 624–649, doi:10.1002/2015JD023927, 2016.

Preusker, R.; Carbajal Henken, C.; Fischer, J. Retrieval of Daytime Total Column Water Vapour from OLCI Measurements over Land
Surfaces. Remote Sens., 13, 932, https://doi.org/10.3390/rs13050932, 2021.

Rue, H.; Held, L.: Gaussian Markov Random Fields (1st Ed.), Chapman and Hall/CRC, New York, https://doi.org/10.1201/9780203492024, 2005.

Saunders, R.; Hocking, J.; Turner, E.; Rayer, P.; Rundle, D.; Brunel, P.; Vidot, J.; Rocquet, P.; Matricardi, M.; Geer, A.; et al. An update on the RTTOV fast radiative transfer model (currently at version 12). Geosci. Model Dev., 11, 1–32, 2018.

Schmetz, J.; Pili, P.; Ratier, A.; Rota, S.; Tjemkes, S. Meteosat Second Generation (MSG): Capabilities and Applications. 2002. Available online: (accessed on 11 December 2021).

Townsend, A.A.: The interpretation of stellar shadow-bands as a consequence of turbulent mixing. Q.J.R. Meteorol. Soc., 91: 1-9. https://doi.org/10.1002/qj.49709138702, 1965.

Tung, K. K., Orlando, W. W.: The k-3 and k-5/3 Energy Spectrum of Atmospheric Turbulence: Quasigeostrophic Two-Level Model Simula-
tion, Journal of the Atmospheric Sciences, 60(6), 824-835, 2003.

Turner, D. D., Ferrare, R. A., Wulfmeyer, V., and Scarino, A. J.: Aircraft evaluation of ground-Based Raman lidar water vapor turbulence profiles in convective mixed layers, J. Atmos. Ocean. Tech., 31, 1078–1088, doi:10.1175/JTECH-D-13-00075.1, 2014.

Wulfmeyer, V., Pal, S., Turner, D. D., and Wagner, E.: Can water vapour Raman lidar resolve profiles of turbulent variables in the convective boundary layer?, Bound.-Lay. Meteorol., 136, 253– 284, doi:10.1007/s10546-010-9494-z, 2010.

Wulfmeyer, V., R. M. Hardesty, D. D. Turner, A. Behrendt, M. P. Cadeddu, P. Di Girolamo, P. Schlüssel, J. Van Baelen, and F. Zus: A review of the remote sensing of lower tropospheric thermodynamic profiles and its indispensable role for the understanding and the simulation of water and energy cycles, Rev. Geophys., 53, 819–895, doi:10.1002/2014RG000476, 2015.

Zuev, V.E.: Laser beams in the atmosphere, New York, Consultant's Bureau, 1982.

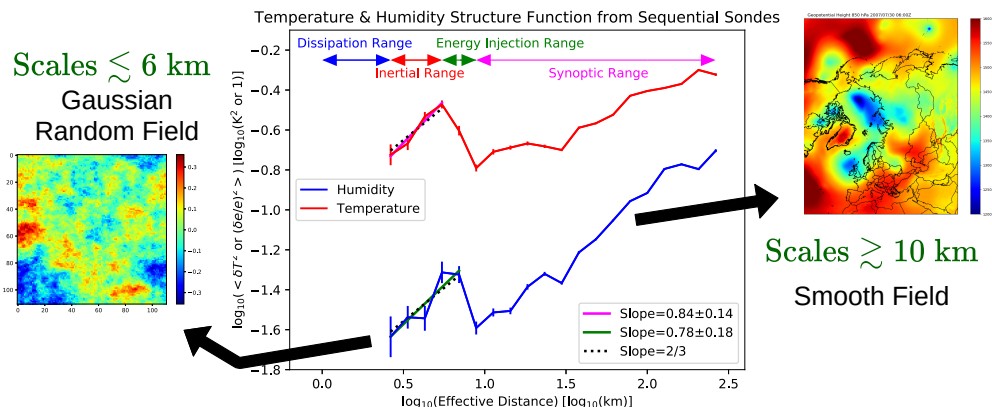

**Figure 1.** Center panel shows an observationally temperature and water vapor structure function (Calbet et al., 2018). Left panel shows a synthetically generated GRF for small spatial scales while the right panel shows a smoothly varying NWP field ta larger spatial scales.



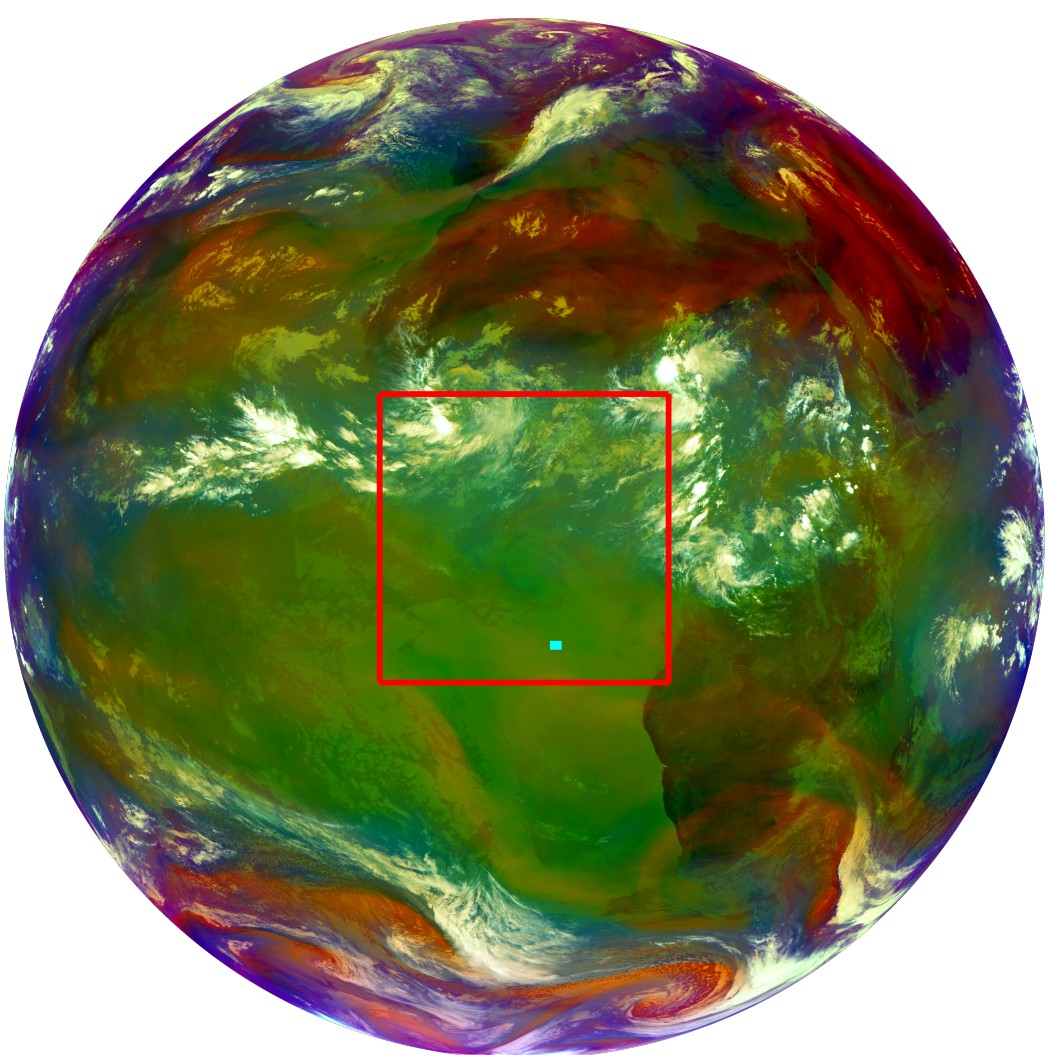

**Figure 2.** SEVIRI/MSG "Airmass RGB" image of the date and time selected (20/08/2021 at 10:00Z). Highlighted in red is the analyzed region. In cyan, the location of the ECMWF profile selected for RTM calculations is shown.

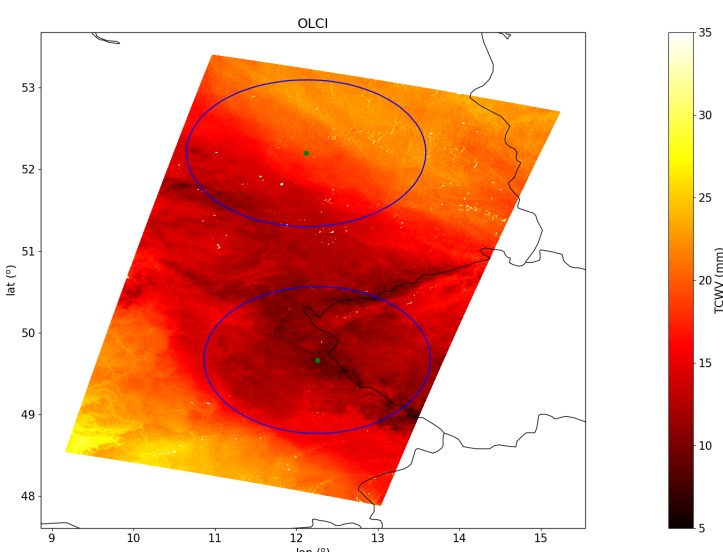

**Figure 3.** OLCI COWa TCWV field from 31/08/2016 at 09:45Z. The blue circles centered on the green dots are regions analyzed in Figs. 5 and 6.

.

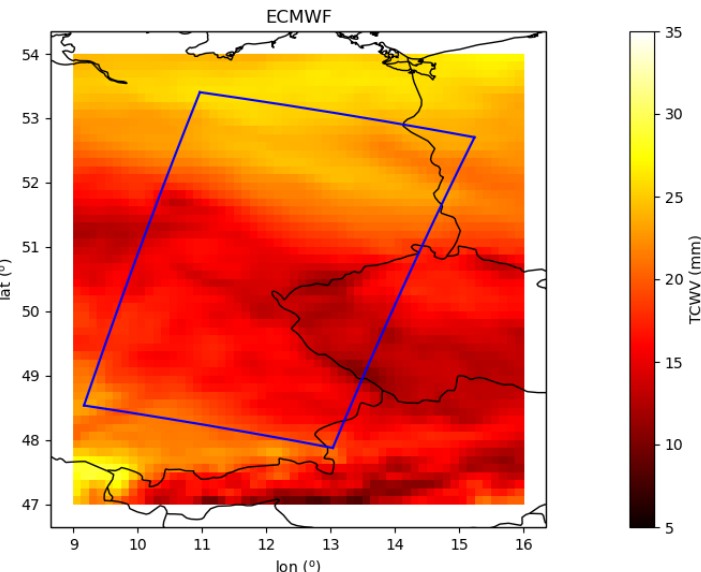

**Figure 4.** ECMWF forecast TCWV field from 31/08/2016 valid at 10:00Z (10 hour step from an analysis at 00Z). The contour of the OLCI observation from Fig. 3 is shown in blue.

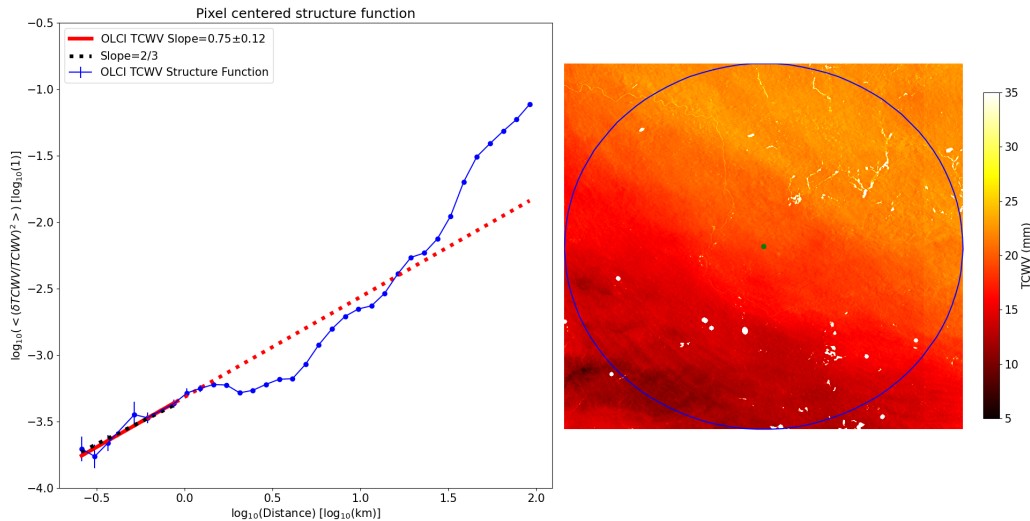

**Figure 5.** Pixel centered structure function for the OLCI COWa TCWV shown in the left plot together with its uncertainty in the vertical axis (blue curve), a linear fit of the points below $10^0$km (red curve) and a two-third slope line (black dots). The structure function covers the area inside the blue circle shown in the right TCWV image which has a radius of 100 km and it is centered in the green dot. This region is a zoom of the top left region highlighted in blue from Fig. 3.



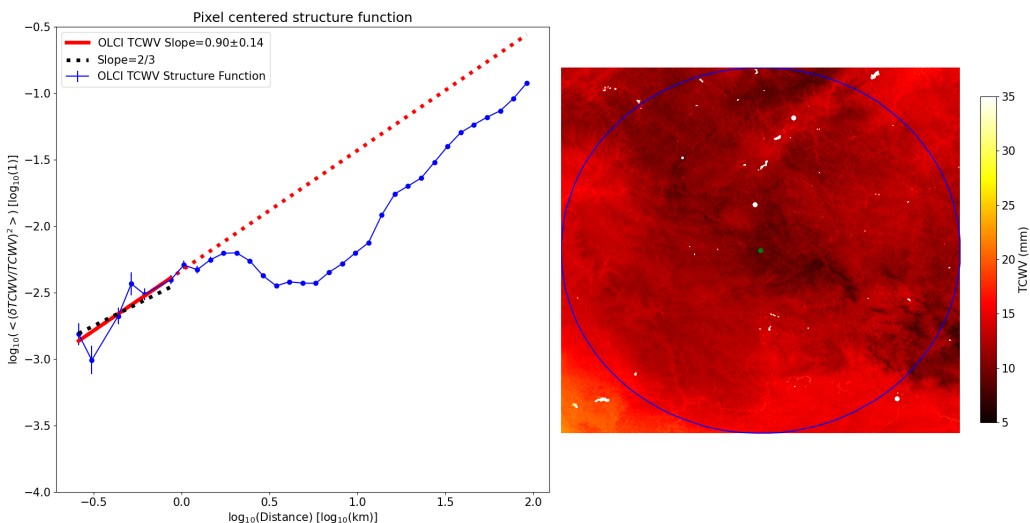

**Figure 6.** Pixel centered structure function for the OLCI COWa TCWV shown in the left plot together with its uncertainty in the vertical axis (blue curve), a linear fit of the points below $10^0$km (red curve) and a two-third slope line (black dots). The structure function covers the area inside the blue circle shown in the right TCWV image which has a radius of 100 km and it is centered in the green dot. This region is a zoom of the bottom right region highlighted in blue from Fig. 3.





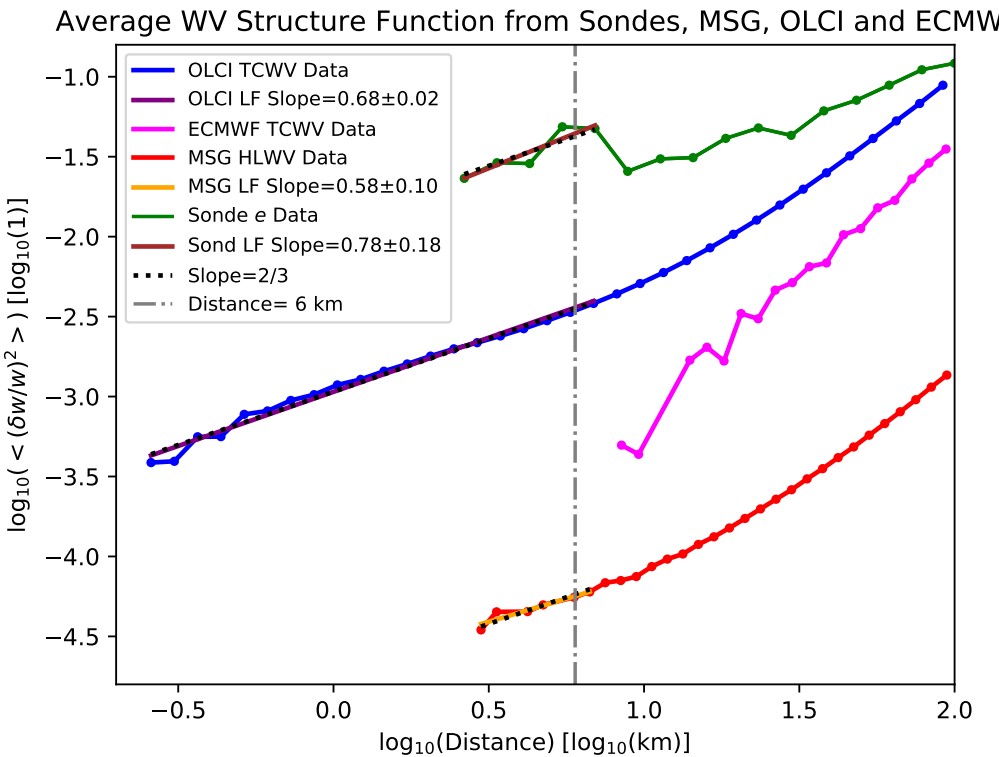

**Figure 7.** Average structure functions for MSG/SEVIRI (red), Sentinel-3/OLCI (blue) and ECMWF forecast (magenta). Also plotted is the plain structure function from radiosondes (green). Linear fits below ∼ 6 km are also shown (orange, purple and brown) together with two-third slope lines (black dots). The distance of 6 km is highlighted as a vertical dash-dotted gray line.

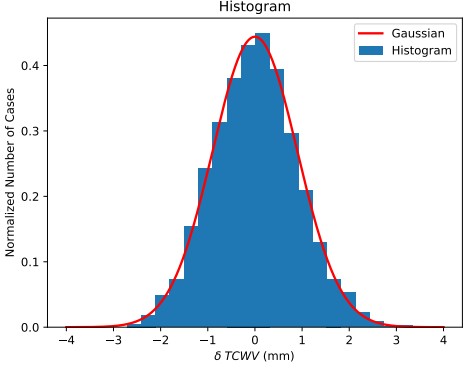

**Figure 8.** Normalized histogram of TCWV differences calculated in boxes within the complete OLCI measurement region (blue) and a Gaussian function with a standard deviation obtained from the data (red).



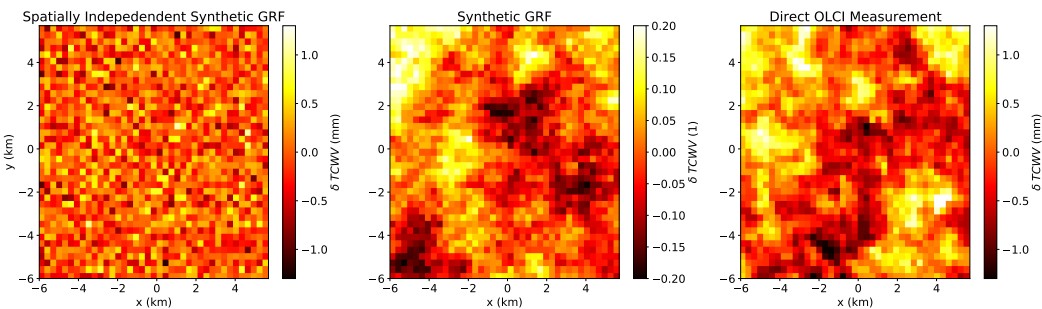

**Figure 9.** Spatially independent ($C(r) = 0$) Gaussian field with standard deviation equal to the OLCI COWa TCWV estimated uncertainty (left), Gaussian random field following the two-thirds law (center), OLCI COWa TCWV difference close up image from Fig 3 centered in $(lon, lat) = (12.2344°, 49.6135°)$ (right).

.