# Peer review of "Horizontal small scale variability of water vapor in the atmosphere: implications for inter-comparison of data from different measuring systems"

_Atmospheric Measurement Techniques, 2022_

## Referee Comment (RC2)

**Small scale variability of water vapor in the atmosphere: implications for inter-comparison of data from different measuring systems**

Xavier Calbert, Cintia Carbajal Henken, Sergo DeSouza-Machado, Bomin Sun, and Tony Reale

This novel study presents a new methodology that demonstrates that the structure of water vapour concentrations at small scales (<6 km) can be approximated by Gaussian Random Fields (GRFs). The authors show that these GRFs have a spatial correlations with a structure function whose gradient is approximately ⅔, which is consistent with Kolmogrov's theory of turbulence. This new methodology is applied to numerical weather forecasting (NWP) fields and a range of real observation systems with different spatial and temporal sampling, with the results discussed in the context of one another. Finally, the results a related to their potential use in Nowcasting.

Overall, I find this study highly relevant to current research and applications within NWP and Nowcasting. Therefore, I recommend this paper suitable for publication after the minor comments I have are addressed.

**Specific Comments:**

- page 2, line 2: reword the sentence "They measure over air regions of the order of linear measures of tens of kilometres." This doesn't read well, could change to something like: e.g. "These instruments measure air mass regions at scales of tens of kilometres."
- page 2, line 6: "mandatory" seems to be the wrong word, do you mean "necessary"?
- page 2, line 18: "below around 6 km", this doesn't read well and could replace around with ~.
- equation 10: does the vapour pressure "e" come in the radiosonde file or has it been calculated? If calculated, what vapour pressure formula was used? Is it consistent with what is used by GRUAN (Hyland and Wexler)? Please state this somewhere in the text.
- page 7, line 27-28: with the images you reference are they L1b files? the term "Level 1.5 ones" reads is a little non-specific. If the files are L1.5 are these fundamentally different to L1b? Therefore, maybe improve the description of the file contents are and put "(L1.5)" at the end of the sentence.
- page 8, lines 24-27: This section needs restructuring, is confusing to someone who is not familiar with NWC SAF workflow. Is the software package the only package NWC SAF produces? A clearer description here is needed.

- page 9, line 7: "a slot" is not an appropriate term here or clear to what is meant. Do you mean a clear sky region? please improve.
- section 3.4: Is this ERA5? ERA Interim? or a specially run forecast at 0.125x0.125 deg run? Current IFS runs in ERA5 are available at 0.25x0.25, is this data just spatially interpolated? PLease clarify and revise this section.
- Methods section: I think that this section could benefit from an algorithm flowchart to help the reader visualise the flow.
- page 12, line 13: replace the phrase "To get a feeling" with something more appropriate, e.g. "To produce a representation"
- section 4.2.3: are all the OLCI regions cloud free? not clear from the text.
- conclusions: Just needs a clear statement on the use or application to Nowcasting. This is done well in the abstract but not so strong here.

---

## Author Response (AR1)

Please note that answers to the reviewers are sentences starting with **

Comment on amt-2022-111
Anonymous Referee #1

Referee comment on "Small scale variability of water vapor in the atmosphere:
implications for inter-comparison of data from different measuring systems" by
Xavier
Calbet et al., Atmos. Meas. Tech. Discuss., https://doi.org/10.5194/amt-2022-
111-RC1,
2022

Review of "Small scale variability of water vapor in the atmosphere:
implications for inter-
comparison of data from different measuring systems", Calbet et al., ATMD 2022.

** Many thanks for the positive review and constructive comments which will for
sure add to the readibility of the paper.

The study and results presented in the manuscript address an interesting
question to
further understand atmospheric processes. The organisation of small-scale WV
structures
is analysed with a small set of satellite data and a global NWP field. The
present study
corroborates previous results obtained with radiosondes, making the experience
that WV
is indeed organised as per the Kolmogorow 2/3-turbulent law.

The paper is found well structured, making a clear didactic introduction of the
turbulent
theory and how it can relate to satellite images. The redaction and explanations
of the
datasets and results are overall of good quality, though inequal in places.

I recommend the publication of the manuscript subject to revision considering
the
comments below:

General:

- try and harmonise the style and level of details provided. It is found too
vague/hasted in
places.
** Tried to refine this in this new version.

- the authors talk about a demonstration with satellite data the WV obey the
2/3-rule.
However the study is made with a rather limited set of satellite observations (1
scene of
SEVIRI, 2 scenes of OLCI) which per se statistically limits the generalisation
of the
conclusions. The figure 9 is a nice result, but is qualitative and one of a
kind. The authors
should acknowledge this, repeat the experience with e.g. other seasons, climate
zones
and/or discuss the representativeness aspects vs generalisation.
** Added in the conclusions "Given the caveat that this is a limited study (not
all seasons or climate zones are covered), the fact that similar results are
obtained from measurements at different layers and from various instruments seem
to proof that this is a universal property which applies at all these ranges."

- the ECMWF global model seems a rather unapropriate choice for this study which
focus

on atmospheric processes below 6km. The authors are encouraged to include regional
models fields, at least in addition to the IFS, and in any case to convolve OLCI data at the
model resolution in an additional analyses. In particular for the region Germany/Czechia,
there should be valuable high-res regional models to work with. The author's findings
would be also certainly of interest to the modellers.
** Absolutely agree, See coments below. We did not have a regional NWP model
available at the time of writing the paper (and still do not have one now).

- the authors should make clearer what are the stakes of characterising the WV
organisation with satellite data. This is briefly touched on in the introduction (e.g. for OBS-
CALC computations) or in the conclusion (e.g. bringing the stochastic components in
weather forecasting), but would deserve some more elaborating. Are there anything
interesting beyond just observing with satellite data that WV is organised after the
2/3-law. Is there a metric about WV variance/turbulence that could be derived from
analysing satellite data and which be supplied to the forecasters to understand better a
given situation?
** Many thanks for this proposal. Added one sentence in the abstract and another
one in the conclusions.
Asbtract: In terms of weather forecasting or nowcasting, the water vapor
variability could be important in estimating the uncertainty of the atmospheric
processes driving convection.

Specific:

Fig.1: Typo "field AT larger spatial scales"
** Changed
Fig.1: would deserve a little more explanations in the caption: what are the geophysical
parameters in those (left and right) fields?
** Extended

P8.L10-19: The strategy is not clear. What is meant by perturbing WV at all levels? Are
you simply training a lienar regression based on synthetic data? And then only applying
the "retrievals" to cloud-free real observations? The concepts behind the approach could
be elaborated more explicitely upfront.
** It is explained,hopefully, better now

P8.L12: "we start by using an atmospheric profile representing all other atmospheric
profiles in the selected region" it is hard to believe that the profile in the cyan tile is
representative of the large red-squared area... it that what is meant? Can you clarify and
elaborate the assumptions behind?
** Please note only a rough first order approximation of the WV is needed. To
make the considerations more understandable to the reader, a simple regression
is used. This is clarified in the tex now.

P9.L11: vague style. You should speak about "uncertainty estimates associated to each
pixel, which on average is expected to lie around 0.33mm".

This by the way sounds extremely ambitious ! Error estimates in OEM greatly depend on
the assumptions made on the background and observation errors, and sometimes may not
be fully representative. Has the uncertainty estimate been validated? It should be referred
here.
** Clarified and reference given in this sentence.

§3.4: Have the authors considered using a regional (convective scale) model? They would
have the potential advantage to resolve more atmospheric processes and at finer scales -
hence be of potential higher relevance for the present study which explores WV structures
on kilometeric if not subkilometric scales. In particular in view of the OLCI study over a
smaller continental portion, this would be very informative. The NWP field from the global
IFS model feels a bit disapointing compared to Fig. 3. In what is it or is it not a limitation
for your study? This should be envisaged or at least explained why regional models are
excluded.
** We agree with the referee. Ideally we would like to compare to a high
resolution regional model. Unfortunately, we do not have direct access to a high
resolution NWP model,so we settled for a global one in order to finish the
paper. This is now clarified in the paper. Hopefully, in the future, this
exercise will be performed. Please also note that this is not a paper in which
NWP models are thoroughly compared to measurements.

P9.25: typo "it is A structure"
** Seems correct as it is. No change.

P10.L18-20: stats do mix-up spatial correlations from very different altitudes, which one
would expect for water-vapour would obey to very different atmospheric processes and
turbulence regimes. How is that an issue for the study and what your are trying to
evaluate regarding the implications of WV spatial structure (beyond the fact that satellite
confirm the 2/3-law expectations)?
** To make this clearea, added in the first paragraph of the conclusiuons "Given
the caveat that this is a limited study (not all seasons or climate zones are
covered), the fact that similar results are obtained from measurements at
different layers and from various instruments seem to proof that this is a
universal property which applies at all these ranges."

P10.L32: "These kind of figures could be reproduced for any other pixel on the complete
OLCI field, showing similar results.". Evasive statement. Do you mean such analyses
WERE successfully performed and showed similar results, but you're displaying just 2
here? Or that you could potentially repeat this approach and you expect finding the same
results? If the former, I suggest working more explicitely and indicating how many such
cases were computed. If the latter, I would avoid what comes across a hypothetical
statement, and would support it by additional experiments.
** They have been made for all pixels. Liekwise for the 2/3 law fitting.
Explained better in the text now.

P11.L17: indeed! a regional model would be a better option (and interesting feed-back to
the modellers too). In what is a 10-km sampling (resolution of phyiscal processes is
typically 2 to 3x coarser) model interesting for your study?
** Agreed. As explained above, no regional high res model was available.

P13.L15-20 The discussion about ECMWF data is too short. The sampling is about 10km
but the resolution of the physical processes is much coarser. How about smoothing OLCI
TCWV field with e.g. a 20-km or 30-km Gaussian running window and repeating the same
analysis and intercomparison? The value of the discussion on the two fields at teir
respective scales is not clear. Using a regional model (with same consideration for
smoothing OLCI to a kilometric Gaussian average) apperas more interesting at first
glance.
** Tried to explain it better now. Also added that further investigation is required. Possibly using a high res NWP model.
* * *
Comment on amt-2022-111
Anonymous Referee #2
Referee comment on "Small scale variability of water vapor in the atmosphere: implications for inter-comparison of data from different measuring systems" by Xavier
Calbet et al., Atmos. Meas. Tech. Discuss., https://doi.org/10.5194/amt-2022-111-RC2,
2022
The comment was uploaded in the form of a supplement: https://amt.copernicus.org/preprints/amt-2022-111/amt-2022-111-RC2-supplement.pdf

Small scale variability of water vapor in
the atmosphere: implications for
inter-comparison of data from different
measuring systems

Xavier Calbert, Cintia Carbajal Henken, Sergo DeSouza-Machado, Bomin Sun, and Tony
Reale

** Many thanks for the positive remarks and the proposed changes which will improve the paper.

This novel study presents a new methodology that demonstrates that the structure of water
vapour concentrations at small scales (<6 km) can be approximated by Gaussian Random
Fields (GRFs). The authors show that these GRFs have a spatial correlations with a
structure function whose gradient is approximately 2/3, which is consistent with Kolmogrov's
theory of turbulence. This new methodology is applied to numerical weather forecasting
(NWP) fields and a range of real observation systems with different spatial and temporal
sampling, with the results discussed in the context of one another. Finally, the results a

related to their potential use in Nowcasting.

Overall, I find this study highly relevant to current research and applications within NWP and
Nowcasting. Therefore, I recommend this paper suitable for publication after the minor
comments I have are addressed.

Specific Comments:

● page 2, line 2: reword the sentence "They measure over air regions of the order of
linear measures of tens of kilometres." This doesn't read well, could change to
something like: e.g. "These instruments measure air mass regions at scales of tens
of kilometres."
** Done

● page 2, line 6: "mandatory" seems to be the wrong word, do you mean "necessary"?
** Done

● page 2, line 18: "below around 6 km", this doesn't read well and could replace around
with ~.
** Done

● equation 10: does the vapour pressure "e'' come in the radiosonde file or has it been
calculated? If calculated, what vapour pressure formula was used? Is it consistent
with what is used by GRUAN (Hyland and Wexler)? Please state this somewhere in
the text.
** Added comment and reference.

● page 7, line 27-28: with the images you reference are they L1b files? the term "Level
1.5 ones" reads is a little non-specific. If the files are L1.5 are these fundamentally
different to L1b? Therefore, maybe improve the description of the file contents are
and put "(L1.5)" at the end of the sentence.
** Done

● page 8, lines 24-27: This section needs restructuring, is confusing to someone who is
not familiar with NWC SAF workflow. Is the software package the only package NWC
SAF produces? A clearer description here is needed.
** Rephrased.

● page 9, line 7: "a slot" is not an appropriate term here or clear to what is meant. Do
you mean a clear sky region? please improve.
** Corrected sentence

● section 3.4: Is this ERA5? ERA Interim? or a specially run forecast at 0.125x0.125
deg run? Current IFS runs in ERA5 are available at 0.25x0.25, is this data just
spatially interpolated? PLease clarify and revise this section.
** It is the operational model obtained at 0.125 x 0.125 degrees. Not ERA5 nor
ERA Interim.
Added operational in the sentence.

● Methods section: I think that this section could benefit from an algorithm

flowchart to
help the reader visualise the flow.
** Agreed with the reviewer that it is not easy to follow the way to do the
calculations. Unfortunately, the calculations involved are difficult to
represent in a flowchart. The explanations seem to be adequate, so it is left as
it is.

● page 12, line 13: replace the phrase "To get a feeling" with something more
appropriate, e.g. "To produce a representation"
** Done

● section 4.2.3: are all the OLCI regions cloud free? not clear from the text.
** Tried to clarify this better in section 3.3 saying that cloudy pixels are
shown as white and also in captions of figures 3, 5 and 6.

● conclusions: Just needs a clear statement on the use or application to
Nowcasting.
This is done well in the abstract but not so strong here.
** Done

---

## Author Response (AR2)

**Answers to the editor 14/11/2022. In bold letters.**

Comment on amt-2022-111
Comments to the author:

- Of course, atmospheric turbulence occurs on the horizontal and vertical scale,
however and importantly, relevant scale length differs strongly in the two
dimensions. And the difference is roughly determined by the slope of the
isentropic surfaces of typically 1:1000 (but in strongly baroclinic conditions
also 1:100). Thus, always specify on which dimension you talk about. The
described variability (scale or scale length) occurs on horizontal scales. Thus,
always add "horizontal" (starting in the abstract), but add "vertical" if you
deal with e.g. LIDAR measurements (e.g. on p.2, l.24f). Writing that there are
"two categories of scale lengths" (p.2, l.24) thus is misleading. Please
improve!
**It has been emphasized in all the text the difference between horizontal and
vertical comparisons. In particular in the abstract and the title. Please, see
track changes for a full view.**

- On p.3, bottom, or in the caption of Figure 1, please explain how the pictures
are inferred or better refer to section 3, e.g. section 3.1. You simply write
"structure function from radiosonde comparisons" (p.3, l.31). Again, write that
you deal here with horizontal scales.
**Paragraph extended and, hopefully, better explained**

- In section 2.2 you come up with many equations that fall from the tree. Please
add relevant citations.
**Included**

- P.6, l.4, wrong grammar.
**Modified**

- P.9, l.30. "Validity time" is a wrong designation. Same on p.13, l.10 and l.13
**Changed to "valid time"**

- I have to admit that I found at least 20 instances where the English spelling
or even grammar is not appropriate or even wrong … it's not my job as editor to
improve this. Please ask a native speaker to check the paper.
**Many parts of the paper re-written. By a native speaker. Hopefully it will
improve somewhat.**

**Many many thanks to the editor for supporting this review and making the final
paper a much more readable and understandable version!!!**